# Effectiveness of Intracerebral Hemorrhage Aspiration with Catheter Insertion: Impact on Hematoma Volume and Symptom Improvement

**DOI:** 10.3390/brainsci15050455

**Published:** 2025-04-26

**Authors:** Jun Kyu Hwang, Na Young Kim, Won Joo Jeong, Chang Ki Jang, Jae Whan Lee, Tae Im Yi, Kwang-Chun Cho

**Affiliations:** 1Department of Neurosurgery, Yongin Severance Hospital, Yonsei University College of Medicine, Yongin 16995, Republic of Korea; nsjk810@yuhs.ac (J.K.H.);; 2Department of Rehabilitation, Yongin Severance Hospital, Yonsei University College of Medicine, Yongin 16995, Republic of Korea; 3Department of Hospital Medicine, Yongin Severance Hospital, Yonsei University College of Medicine, Yongin 16995, Republic of Korea

**Keywords:** catheter-based drainage, endoscopic, endoscopic intracerebral hemorrhage evacuation, intracerebral hemorrhage, minimally invasive

## Abstract

**Background:** Catheter insertion is the most commonly used method for treating intracerebral hemorrhage (ICH). Simultaneous hematoma aspiration allows for faster decompression than catheter insertion alone. **Methods:** Between March 2020 and Apri1 2024, 49 patients (25 men and 24 women) with ICH underwent ICH catheter insertion. Thirty-two patients (Group A) underwent intraoperative aspiration simultaneously with ICH catheter insertion. The other seventeen patients underwent catheter insertion alone (Group B). Four patients in Group A (12.5%) and two patients in Group B (11.8%) died within one month after surgery. Consequently, a total of 43 patients were included in the final analysis (Group A-1 vs. Group B-1). The Glasgow coma scale (GCS) score and muscle strength of the four extremities in both groups were compared 2 weeks later (first period) and on the date of discharge (second period). Patients in Group A-1 were discharged 5 days earlier than those in Group B-1. (average 49 ± 20 vs. 54 ± 31 days). **Results:** In Group A, the preoperative ICH volume was 66.2 ± 28.8 mL, and the median aspiration volume was 30 ± 19.6 mL. The preoperative ICH volume was 55.9 ± 22.2 mL in group B. Intraoperative ICH aspiration (Group A-1) significantly improved motor scores during the second period (*p* = 0.001). It also showed the trend toward improved GCS scores during the first period (*p* = 0.095) and the second period (*p* = 0.069). **Conclusions:** Compared to ICH catheter insertion alone, additional intraoperative ICH aspiration resulted in greater motor improvement at 7 weeks postoperatively (*p* = 0.004). It also showed a trend toward greater improvement in the GCS scores (*p* = 0.12).

## 1. Introduction

Spontaneous intracerebral hemorrhage (ICH) accounts for 8% to 30% of all stroke cases. It is the most severe type, with a 1-year survival rate below 50% and a mortality rate two to six times higher than that of ischemic stroke [1,2,3,4]. The annual incidence is approximately 20 cases per 100,000 population [5]; however, no standard surgical procedure has proven effective for ICH. Furthermore, there is ongoing debate regarding treatment approaches, which include craniotomy for hematoma removal, endoscopic evacuation, aspiration, catheter insertion, and conservative care [4,6,7,8,9,10,11].

Theoretically, surgical hematoma removal should effectively manage intracranial pressure (ICP) and reduce the neurotoxic effects of blood breakdown products [12]. Additionally, craniotomy is considered the most effective method for volumetric hematoma removal and achieving hemostasis. However, extensive surgical manipulation often leads to secondary brain tissue injury. James et al. indicated a median age of 75 years among patients with ICH based on a study of 192,826 cases [13]. Additionally, Nobleza reported that approximately one-third of patients with ICH were taking anticoagulants, antiplatelets, or both types of medications [14]. Advanced age and anticoagulant use in patients with ICH present challenges for major surgery. However, conservative therapy alone cannot rapidly reduce ICP, and perihematoma edema may worsen, often necessitating surgical intervention.

As a compromise between these two approaches, minimally invasive surgical techniques, such as endoscopic evacuation, aspiration, and catheter insertion, are increasingly preferred, with several studies conducted to enhance their effectiveness [12,15,16,17,18]. Among these, catheter insertion alone is widely adopted as it minimizes re-bleeding while maximizing the benefits of minimally invasive techniques. For these reasons, neurosurgeons at our institution have performed ICH catheter insertions without additional techniques. However, the first author of this study performed simultaneous ICH aspiration for faster decompression and, in some cases, used an endoscope for additional irrigation. A retrospective analysis was conducted to evaluate the effect of additional ICH aspiration between the two groups.

## 2. Methods

### 2.1. Patient Selection

Between March 2020 and April 2024, a total of 525 patients were admitted to the Department of Neurosurgery with a diagnosis of spontaneous ICH. Among them, 100 patients (19.0%) underwent surgical intervention for ICH removal. Most of these patients were admitted via the emergency room, and surgical intervention was actively performed in cases presenting with a mental status of deep drowsiness or worse or with evident hemiparesis. Notably, surgery was indicated when there was no neurological improvement despite the immediate initiation of mannitol administration, hyperosmolar therapy, and pain control prior to the procedure.

Among them, 49 patients (25 men and 24 women) underwent ICH catheter insertion at our institution. Of these, 32 patients underwent ICH catheter insertion with intentional aspiration (Group A) and initial volume measurement. Volume was manually calculated using the ZeTTA picture archiving and communication system (PACS) (Taeyong Soft Co., Ltd., Gwacheon, Gyeonggi Province, Republic of Korea). Twelve patients in Group A (37.5%) underwent additional endoscopic irrigation through the same trajectory as the ICH catheter. Unlike endoscopic evacuation procedures at other institutions, our approach used a ventriculoscope in a limited manner, prioritizing re-bleeding prevention while irrigating only non-viscous fluid. Seventeen patients (Group B) underwent ICH catheter insertion without intentional aspiration or endoscopic irrigation. One or two ICH catheters were inserted, with two being more frequently used in Group B (*p* = 0.008) (Table 1). Among the 49 patients who underwent ICH catheter insertion, 6 (12.2%) patients died within one month postoperatively due to complications. Outcomes according to surgical method were analyzed in 28 patients from Group A (Group A-1) and 15 patients from Group B (Group B-1) (Figure 1).

### 2.2. Intraoperative ICH Aspiration

Before surgery, ICH volume and ventricular opening were assessed using brain computed tomography (CT) images using the PACS. Under general endotracheal anesthesia, entry points and trajectories were planned using the NAV3i surgical navigation system (Stryker, Kalamazoo, MI, USA) with horseshoe head fixation.

A 2–3 cm skin incision was made in each case, followed by burr-hole creation with a high-speed drill. Thereafter, a cross-shaped dural incision was made, and an external ventricular drainage (EVD) catheter (12 French, double lumen; Yushin Medical Co., Ltd., Bucheon, Gyeonggi Province, Republic of Korea) was inserted along the planned trajectory as an ICH catheter. Upon detecting hemorrhage drainage through the lumen, the catheter was clamped with mosquito forceps to accurately measure the opening pressure and aspirated ICH volume.

After clamping the ICH catheter, the proximal sensor of a pressure monitor (Codman^®^ ICP Express^®^ MicroSensor; Johnson & Johnson, Raynham, MA, USA) was inserted at the same site and depth. After measuring the initial ICH pressure, 10 mL syringes were used for active aspiration. After hematoma aspiration, the catheter tip was repositioned within a 2 cm radius for additional aspiration. After three or four passes, active aspiration ceased upon confirming ICP normalization (<15 mmHg). The ICH catheter remained as planned, and the incisions were closed in layers using 3-0 Vicryl and 4-0 Vicryl Rapid sutures(B. Braun Melsungen AG, Melsungen, Germany). Twelve patients in Group A (37.5%) underwent additional irrigation using a rigid ventriculoscope (little LOTTA System, Karl storz, Tuttlingen, Germany) (Figure 2). During irrigation, Hartmann’s solution was used with a pressure maintained below 50 mmHg.

### 2.3. Thrombolytic Agent Administration Through the ICH Catheter

Twenty-two patients (68.7%) in Group A and ten (58.8%) in Group B received thrombolytic agents through the ICH catheter (*p* = 0.299). One milligram of alteplase (Actilyse^®^, Boehringer Ingelheim, Inc., Bracknell, Berkshire, UK) was injected through the small lumen of the catheter [19]. Following the method from a previous phase 2 trial, alteplase was administered if the postoperative ICH volume exceeded 20 mL [20]. However, the frequency of its use was conservative, considering the risk of re-bleeding. Alteplase was first infused 6 h after surgery due to the high risk of re-bleeding [21]. Patients who received alteplase were infused once or twice, except in one case in Group B (thrice), at 24 h intervals, with a daily CT scan during each infusion. ICP was measured hourly through the catheter. To prevent over-drainage and promote natural drainage, the catheter height ranged from 0 to 10 cm, with squeezing performed if drainage was interrupted by the hematoma.

### 2.4. Glasgow Coma Scale (GCS) and Medical Research Council (MRC) Scale Score Analysis

All patients received standard medical management following the guidelines for treating spontaneous ICH in adults, as published by the American Heart Association/American Stroke Association Stroke Council, High Blood Pressure Research Council, and Quality of Care and Outcomes in Research Interdisciplinary Working Group [22]. The GCS and sum of the Medical Research Council scale (sMRCS) scores for the extremities were assessed and compared between the two groups [23,24,25]. The sMRCS scores of the four extremities were used to analyze motor function [26]. To maintain blinding, scores were obtained from nursing records, where staff were unaware of the surgical differences. Neurological examination results at 2 weeks (first period) and on the day of discharge (second period) were retrospectively reviewed. Four patients in Group A (12.5%) and two in Group B (11.8%) were excluded from this serial comparison due to death within 1 month postoperatively. All six patients are believed to have died from complications of ICH rather than from surgical complications. Discharge (second period) occurred at 49 ± 20 days for Group A and 54 ± 31 days for Group B. Although no significant difference (*p* = 0.393) was observed, the Group A patients were discharged 5 days earlier than the Group B patients, suggesting earlier improvement in Group A.

Postoperative care was provided in the neurosurgery department, followed by early rehabilitation if appropriate. One week postoperatively, most patients received sedative medication for adequate ICP control. Therefore, 2 weeks postoperatively was considered the most suitable period for comparing immediate neurological symptoms without sedation. After the intensive care unit (ICU) stay, patients were transferred to the ward and continued care in collaboration with the hospital medicine department. After postoperative care, patients were transferred to the department of rehabilitation.

### 2.5. Statistical Analysis

Linear mixed models were used to calculate the beta coefficient and 95% confidence interval for improved the GCS score and motor function. Data on the baseline GCS score, sMRCS score, ICH aspiration, sex, age, location, initial ICH volume, surgery type, presence of intraventricular hemorrhage (IVH), number of ICH and EVD catheters, medical history, time from symptom onset to surgery, postoperative alteplase use, and preoperative anticoagulant use were considered as independent variables in the linear mixed model analysis. Univariable analysis was first conducted on these independent variables, and those with *p*-values below 0.05 were included in the multivariable analysis. In the multivariable analysis, we assessed the interaction effect of ICH aspiration and time, calculating least-squares mean values to compare improvements in the GCS scores and motor function postoperatively. A two-sided *p*-value below 0.05 was considered statistically significant. All statistical analyses were performed using the R software (version 4.4.1; R Foundation for Statistical Computing, Vienna, Austria).

### 2.6. Data Availability Statement

The datasets analyzed during the current study are not publicly available due to patient confidentiality and ethical restrictions imposed by the Institutional Review Board (IRB) but are available from the corresponding author upon reasonable request.

## 3. Results

The average initial ICH volume was 66.2 ± 29.8 mL in Group A and 59.9 ± 22.2 mL in Group B, indicating that Group A had a slightly larger volume. In Group A, initial ICP was measured during ICH aspiration and was 25.0 ± 18.5 mmHg. A total of 37.2 ± 19.5 mL of hematoma was aspirated. After completing both aspiration and irrigation, the ICP decreased to 3.4 ± 9.2 mmHg, which is within the normal intracranial pressure range. In Group A, one patient (3.1%) required two ICH catheters, whereas in Group B, six patients (35.3%) underwent insertion of two catheters (*p* < 0.008). Retrospectively, this suggests that Group B relied more on the insertion of multiple catheters rather than aspiration. The male-to-female ratio was approximately 1:1 in both groups. In Group A, 22 patients (68.7%), and in Group B, 10 patients (58.8%) received alteplase, with most patients receiving it only once or twice. In Group B, one patient required a third administration due to residual ICH (>20 mL) after two doses. Twelve patients (37.5%) in Group A underwent additional endoscopic irrigation, though this was not a significant factor (*p* = 0.665, *p* = 0.284). No significant differences were observed between the two groups regarding sex, age, location, presence of IVH, EVD insertion, medical history, time from symptom onset to surgery, or preoperative anticoagulant use (Table 1).

In the univariable analysis, the GCS and sMRC scale scores improved significantly during the first period, with further improvements observed in the second period (*p* < 0.001). Initial ICH volume, presence of IVH, EVD insertion, chronic kidney disease, and time from symptom onset to surgery were all significantly associated with the GCS or sMRCS scores (*p* < 0.05) (Table 2).

In the multivariable analysis, motor function improved significantly during the second period (*p* = 0.001). The GCS scores showed a trend of improvement in the first (*p* = 0.095) and second periods (*p* = 0.069). Preoperative ICH volume significantly negatively affected the GCS scores (Table 3).

We calculated the least-squares mean values from the multivariable analysis model of the two groups to analyze these findings further. Both the GCS and sMRCS scores showed significant improvements during both periods compared with the baseline (*p* < 0.05). When comparing the two groups, Group A showed a trend toward greater improvement in the GCS scores (*p* = 0.120), whereas the sMRCS scores demonstrated significantly greater improvement (*p* = 0.004) (Table 4).

Both groups showed gradual postoperative improvement in the GCS scores and motor function. However, when comparing the least-squares mean values from the multivariable analysis model, Group A showed faster improvement in both measures. The difference was particularly pronounced in motor function (Figure 3, Table 4).

## 4. Discussion

Some studies have reported that operative management of ICH does not influence long-term clinical symptom improvement [10,27,28]. The International Surgical Trial in Intracerebral Hemorrhage (STICH) and STICH II found no clinical benefit of early surgical evacuation of intraparenchymal hematoma over medical management with delayed surgery, if necessary, in patients with spontaneous ICH [6,9,10]. However, the STICH trials may not be generalizable due to high crossover rates from medical to surgical management. The Minimally Invasive Surgery Plus rt-PA for Intracerebral Hemorrhage Evacuation (MISTIE) III trial, a large randomized, controlled, open-label, phase 3 study combined with a meta-analysis, compared minimally invasive surgery (MIS) with thrombolysis against conservative management. Although safe, MIS did not confer long-term functional benefits over conservative management [8].

Nevertheless, these trials have limitations. Operative intervention was performed in approximately 25% of patients in the initial conservative treatment group in the STICH and STICH II trials. The MISTIE III trial used the modified Rankin Scale and extended Glasgow outcome scale to assess surgical outcomes; however, these measures do not comprehensively capture ICH symptoms. The lack of blinding may have also introduced undertreatment or overtreatment bias in treatment allocation.

Conversely, studies supporting the superiority of surgical treatment—particularly MIS—continue to be reported. The Early Minimally Invasive Removal of Intracerebral Hemorrhage (ENRICH) trial was a randomized controlled study that compared MIS with medical management in 300 patients with ICH located in the anterior basal ganglia or cortex, with hematoma volumes between 30 and 80 mL. At 180 days, the surgical group demonstrated better functional outcomes, and the 30-day mortality rate was lower in the surgical group (9.3%) compared to the control group (18.0%) [17]. The Minimally Invasive Surgeries for Spontaneous Hypertensive Intracerebral Hemorrhage (MISICH) trial, conducted across 16 centers in China, evaluated 733 patients with hypertensive subcortical ICH and compared outcomes among endoscopic surgery, stereotactic aspiration, and craniotomy. Endoscopic surgery and stereotactic aspiration showed better functional outcomes at 6 months compared to craniotomy, with the benefits of MIS being particularly pronounced in patients with hemorrhages involving the basal ganglia and thalamus [18].

Surgical hematoma removal can be lifesaving for patients with a GCS score between 9 and 12 (moderate quality, weak recommendation) [11]. MIS is effective when a major procedure is unfeasible despite a large ICH. Conversely, rapid perilesional decompression of the hematoma—unattainable through conservative care alone—can be achieved with MIS. Recent advancements in navigation systems and endoscopic techniques have greatly improved surgical outcomes. In this study, ICH catheter insertion accuracy was enhanced using a navigation system, while endoscopy enabled direct visualization during additional irrigation when necessary. Given the rapid reduction in ICP and elimination of substantial neurotoxic products, medication alone presents clear disadvantages compared with surgical hematoma removal. A systematic review and meta-analysis found MIS to be superior to craniotomy and conservative management in reducing mortality, dependence, and complications [29,30,31,32,33]. It also facilitated early rehabilitation and reduced intraoperative bleeding, ICU stays, and complications such as pneumonia, urinary tract infections, and pressure sores [4].

In Group A, ICH aspiration enabled a rapid reduction in pressure and hematoma volume within minutes. Some damaged brain cortex was also aspirated along with the hematoma; however, the priority was rapid pressure reduction.

There are reports suggesting that immediate surgery may not always be breneficial due to the risk of re-bleeding [10,34,35,36]. However, we focused more on achieving a more rapid reduction in intracranial pressure, and no intraoperative re-bleeding occurred in this study. Nevertheless, we believe that the risk of bleeding may be higher compared to cases with catheter insertion alone. To reduce the risk of re-bleeding, hyperosmolar therapy and pain control were initiated immediately upon the patient’s arrival at the emergency room. Aspiration was performed using a 10 mL syringe to apply only mild pressure, and irrigation was conducted at pressures not exceeding 50 mmHg. Upon reviewing the immediate postoperative brain CT scans, no re-bleeding was observed following endoscopic irrigation in this study. However, due to the limited surgical field and the presence of hematoma-mixed fluid, it is considered very difficult to accurately identify intraoperative re-bleeding.

Our analysis of the GCS and sMRCS scores showed that this technique provided greater benefits compared with catheter insertion alone. Significant sMRCS score improvement occurred in the first period, with continued progress throughout the second period. Intraoperative ICH aspiration likely contributed to the initial improvement, facilitating early rehabilitation and significant motor function improvements in the second period. Despite the blinded analysis, the Group A patients were discharged 5 days earlier than the Group B patients. The reduced hospitalization duration suggests a more rapid recovery of clinical symptoms.

Multiple studies have reported that early post-stroke rehabilitation enhances functional recovery [37,38,39,40]. In the MISTIE III trial, longitudinal data indicated that functional independence at 30 days increased four-fold by 365 days [8]. Motor function improvements during the second period may ultimately influence long-term rehabilitation outcomes. 

In Group A-1, 21 patients (75.0%) were followed-up at the outpatient clinic for an average of 7.9 ± 4.9 months, with an sMRCS score of 14.7 ± 4.3. In Group B-1, 11 patients (73.3%) were followed-up at the outpatient clinic for an average of 6.7 ± 13.7 months, with an sMRCS score of 12.1 ± 4.5. An independent samples *t*-test was conducted to compare the mean values between Group A and Group B. The analysis revealed no statistically significant difference between the two groups (t = 1.88, *p* = 0.076). Although the sample size was limited and statistical significance was not achieved, the improvement in sMRCS observed in the second period was maintained approximately 7 months later, and a trend toward a difference between the two groups was observed (*p* = 0.076). Despite the absence of documented GCS scores, alert patients were less likely to be lost to follow-up.

Inadequate hematoma removal via MIS often requires a postoperative thrombolytic agent. Thrombolytic agents for ICH pose a high risk of re-bleeding and remain limited due to unstandardized dosages and various drug-related side effects, such as pro-edema, pro-inflammation, and pro-neurotoxicity [19,41,42,43]. Notably, no major re-bleeding events occurred in our study. However, undetected re-bleeds likely occurred, considering the immediate increase in ICH drainage following thrombolytic agent injection. Following thrombolytic agent administration, repeated squeezing was performed through the ICH catheter.

ICP elevation is a result of the hematoma mass effect and perihematomal edema [44]. ICH aspiration can simultaneously remove the hematoma and damaged brain cortex matter, with its effectiveness demonstrated by normalizing ICH pressure after aspiration. Aspiration during surgery was terminated only after confirming ICP normalization through pressure monitoring. However, uncontrolled brain edema must also be treated with postoperative medical management and thrombolytic agents.

The study has certain limitations, including its retrospective design, small sample size, and lack of randomization, which may introduce bias and reduce comparability. The short follow-up period and exclusion of deceased patients limit the assessment of long-term outcomes and risks. As a single-center study, its generalizability is restricted, and reliance on nursing records for assessments may introduce measurement bias. In the future, it would be preferable to conduct a prospective cohort study with a larger patient population for more robust analysis.

## 5. Conclusions

Intraoperative ICH aspiration rapidly reduces perihematoma pressure, leading to greater symptom improvement, compared with implanting an ICH catheter alone. This led to earlier rehabilitation and greater motor improvement within 2 weeks, with continued progress over 7 weeks.

## Figures and Tables

**Figure 1 brainsci-15-00455-f001:**
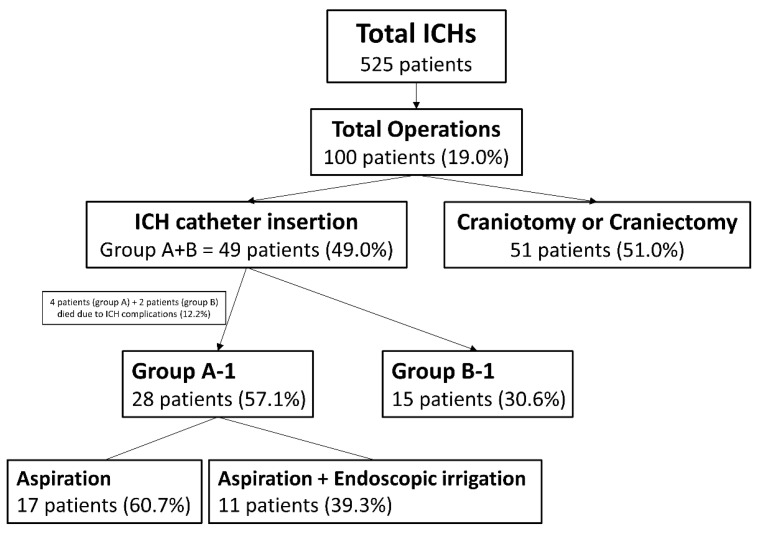
Classification of patients according to treatment methods.

**Figure 2 brainsci-15-00455-f002:**
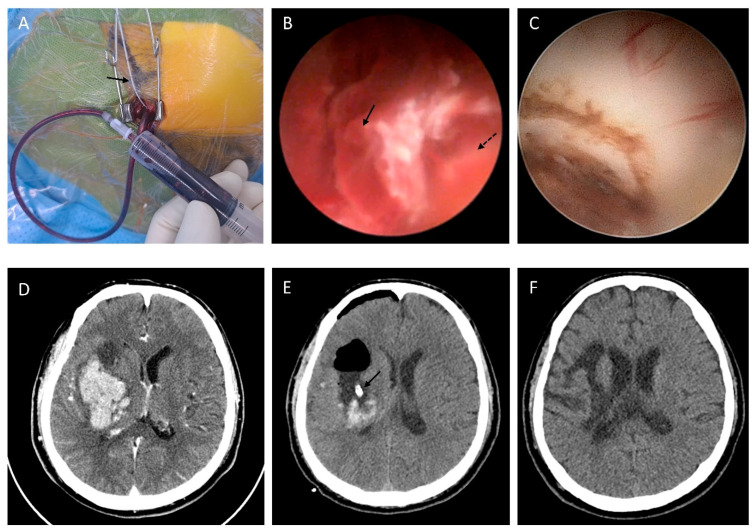
A 51-year-old male patient with a right basal ganglia ICH measuring 74.5 mL. (**A**) A 2 cm skin incision was made along a predetermined trajectory, followed by burr-hole creation and insertion of a 12-Fr EVD catheter. Aspiration was performed using a 10 mL syringe, and ICP was simultaneously monitored at the same site (solid arrow). (**B**) Endoscopic view showing the hematoma (solid arrow) and the brain margin (dashed arrow). (**C**) Endoscopic visualization of the brain tissue margin. (**D**) Preoperative CT scan. (**E**) Immediate postoperative CT scan showing accurate catheter placement according to the planned trajectory (solid arrow). (**F**) Follow-up CT scan taken two months after surgery demonstrating complete hematoma resolution and resolution of brain edema.

**Figure 3 brainsci-15-00455-f003:**
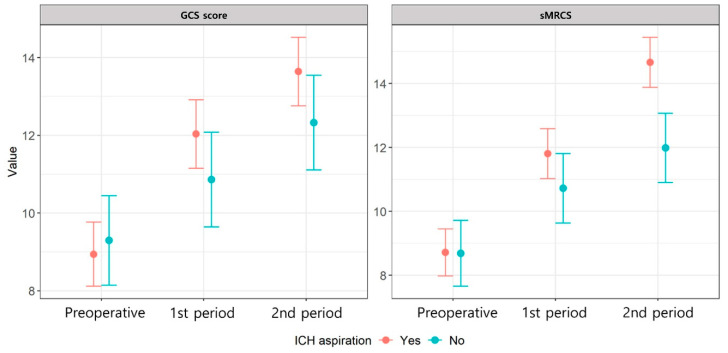
Comparison of least-squares means from the multivariable analysis model of the two groups. ICH, intracerebral hemorrhage; GCS, Glasgow coma scale.

**Table 1 brainsci-15-00455-t001:** Patient characteristics (n = 49).

Characteristic	ICH Aspiration		*p*-Value
	Yes (n = 32)	No (n = 17)	
Sex			0.917
Male	17 (53.1%)	8 (47.1%)	
Female	15 (46.9%)	9 (52.9%)	
Age	61.13 ± 14.89	62.24 ± 19.45	0.839
Location			0.401
Left basal ganglia	8 (25.0%)	8 (47.1%)	
Right basal ganglia	11 (34.4%)	5 (29.4%)	
Thalamus	1 (3.1%)	1 (5.9%)	
Posterior fossa	3 (9.4%)	0 (0.0%)	
Subcortical	9 (28.1%)	3 (17.6%)	
ICH volume (mL)	66.18 ± 29.81	59.92 ± 22.21	0.411
Surgery			<0.001
Catheter insertion only	0 (0.0%)	17 (100.0%)	
ICH aspiration	20 (62.5%)	0 (0.0%)	
Endoscopic irrigation after aspiration	12 (37.5%)	0 (0.0%)	
IVH, yes	18 (56.3%)	5 (29.4%)	0.136
Number of ICH catheters			**0.008** *****
1	31 (96.9%)	11 (64.7%)	
2	1 (3.1%)	6 (35.3%)	
EVD insertion	1 (3.1%)	2 (11.8%)	0.565
Past medical history			
Heart disease	18 (56.3%)	10 (58.8%)	1.000
Diabetes mellitus	5 (15.6%)	2 (11.8%)	1.000
Liver disease	1 (3.1%)	0 (0.0%)	1.000
CKD	2 (6.3%)	2 (11.8%)	0.902
Pulmonary disease	0 (0.0%)	1 (5.9%)	0.745
Cerebrovascular disease	3 (9.4%)	3 (17.6%)	0.702
Time from symptom to surgery			0.740
<4 h	8 (25.0%)	5 (29.4%)	
4–12 h	13 (40.6%)	5 (29.4%)	
≥12 h	11 (34.4%)	7 (41.2%)	
Postoperative actylase			0.371
0	10 (31.3%)	7 (41.2%)	
1	9 (28.1%)	5 (29.4%)	
2	13 (40.6%)	4 (23.5%)	
3	0 (0.0%)	1 (5.9%)	
Anticoagulant use (preoperative)	5 (15.6%)	4 (23.5%)	0.770

ICH, intracerebral hemorrhage; IVH, intraventricular hemorrhage; EVD, external ventricular drainage; CKD, chronic kidney disease, * *p* < 0.05.

**Table 2 brainsci-15-00455-t002:** Univariate analysis of Glasgow coma scale scores and sum of the Medical Research Council scale scores.

Variable	GCS Score			sMRCS Score		
	Coefficient	95% CI	*p*-Value	Coefficient	95% CI	*p*-Value
ICH aspiration						
No (Group B-1)	Reference			Reference		
Yes (Group A-1)	0.255	−1.884, 2.394	0.811	−0.985	−3.931, 1.960	0.504
Time						
Baseline	Reference			Reference		
First period	2.654	1.766, 3.542	<0.001	2.779	1.960, 3.597	<0.001
Second period	4.212	3.324, 5.101	<0.001	5.081	4.262, 5.900	<0.001
Baseline value	0.792	0.635, 0.948	<0.001	0.922	0.811, 1.033	<0.001
Sex						
Male	Reference			Reference		
Female	−0.754	−2.790, 1.282	0.459	0.984	−1.826, 3.794	0.484
Age	−0.040	−0.102, 0.022	0.202	−0.039	−0.126, 0.048	0.368
Location						
Left basal ganglia	Reference			Reference		
Right basal ganglia	0.426	−2.162, 3.013	0.741	0.131	−3.454, 3.717	0.941
Thalamus	−2.873	−8.210, 2.465	0.283	−1.150	−8.584, 6.284	0.756
Posterior fossa	−1.030	−5.713, 3.654	0.660	−0.252	−6.680, 6.176	0.937
Subcortical	0.577	−2.203, 3.356	0.677	1.925	−1.931, 5.781	0.319
ICH volume	−0.047	−0.081, −0.014	0.007	−0.069	−0.115, −0.023	0.004
Surgery						
No	Reference			Reference		
Catheter	0.755	−1.533, 3.042	0.509	−0.451	−3.606, 2.704	0.775
Endoscopy	−0.553	−3.111, 2.004	0.665	−1.881	−5.366, 1.605	0.284
IVH						
No	Reference			Reference		
Yes	−2.673	−4.592, −0.754	0.007	−4.132	−6.714, −1.550	0.002
Number of catheters						
1	Reference			Reference		
2	−1.091	−3.977, 1.795	0.450	−0.805	−4.808, 3.197	0.687
EVD						
No	Reference			Reference		
Yes	−5.012	−9.141, −0.883	0.018	−3.671	−9.554, 2.212	0.216
Heart disease						
No	Reference			Reference		
Yes	−1.355	−3.392, 0.683	0.187	−2.176	−4.966, 0.614	0.123
DM						
No	Reference			Reference		
Yes	0.651	−2.365, 3.668	0.666	−0.399	−4.514, 3.717	0.846
Liver disease						
No	Reference			Reference		
Yes	−3.049	−10.029, 3.930	0.382	−6.601	−16.153, 2.950	0.170
CKD						
No	Reference			Reference		
Yes	−3.546	−7.564, 0.472	0.083	−6.063	−11.290, −0.836	0.024 *
Pulmonary disease						
No	Reference			Reference		
Yes	−4.758	−11.634, 2.117	0.169	−5.919	−15.515, 3.678	0.220
Cerebrovascular disease						
No	Reference			Reference		
Yes	0.407	−2.646, 3.460	0.789	3.740	−0.339, 7.819	0.071
Time from symptom to surgery						
<4 h	Reference			Reference		
4–12 h	2.142	−0.250, 4.535	0.078	1.710	−1.492, 4.913	0.289
≥12 h	2.500	0.258, 4.742	0.029 *	3.765	0.908, 6.622	0.010 *
Postoperative alteplase						
0	Reference			Reference		
1	1.057	−1.109, 3.222	0.333	1.229	−1.635, 4.094	0.396
2	−1.647	−3.912, 0.619	0.150	−2.197	−5.369, 0.975	0.170
3	−6.045	−12.741, 0.651	0.075	−5.649	−15.244, 3.946	0.241
Preoperative anticoagulant						
No	Reference			Reference		
Yes	−1.158	−3.803, 1.487	0.382	−1.195	−4.836, 2.446	0.512

GCS, Glasgow coma scale; sMRCS, sum of the Medical Research Council scale; CI, confidence interval; ICH, intracerebral hemorrhage; IVH, intraventricular hemorrhage; EVD, external ventricular drainage; DM, diabetes mellitus; CKD, chronic kidney disease, * *p* < 0.05.

**Table 3 brainsci-15-00455-t003:** Multivariate analysis of Glasgow coma scale scores and sum of the Medical Research Council scale scores.

Variable	GCS Score			sMRCS Score		
	Coefficient	95% CI	*p*-Value	Coefficient	95% CI	*p*-Value
ICH aspiration						
No (Group B-1)	Reference			Reference		
Yes (Group A-1)	−0.354	−1.798, 1.089	0.627	0.031	−1.265, 1.327	0.962
Time						
Baseline	Reference			Reference		
First period	1.565	0.111, 3.020	0.035	2.033	0.759, 3.308	0.002
Second period	3.032	1.578, 4.486	<0.001	3.300	2.026, 4.574	<0.001
ICH aspiration × time						
ICH aspiration, Yes, 2 weeks after surgery	1.529	−0.271, 3.329	0.095	1.054	−0.523, 2.631	0.188
ICH aspiration, Yes, 2 months after surgery	1.670	−0.130, 3.470	0.069	2.644	1.067, 4.221	0.001 *
Baseline value	0.680	0.515, 0.845	<0.001	0.867	0.747, 0.988	<0.001
Preoperative ICH volume	−0.023	−0.042, −0.004	0.018 *	−0.011	−0.029, 0.007	0.216
IVH						
No	Reference			Reference		
Yes	0.152	−1.039, 1.344	0.798	−0.606	−1.648, 0.435	0.246
EVD						
No	Reference			Reference		
Yes	−0.765	−3.261, 1.732	0.541	0.728	−1.446, 2.902	0.504
CKD						
No	Reference			Reference		
Yes	−0.841	−3.120, 1.438	0.464	−0.765	−2.830, 1.301	0.462
Time from symptom to surgery						
<4 h	Reference			Reference		
4–12 h	0.795	−0.477, 2.066	0.215	−0.823	−1.982, 0.337	0.160
≥12 h	0.961	−0.271, 2.192	0.124	−0.081	−1.258, 1.096	0.891

GCS, Glasgow coma scale; sMRCS, sum of the Medical Research Council scale; CI, confidence interval; ICH, intracerebral hemorrhage; IVH, intraventricular hemorrhage; EVD, external ventricular drainage; CKD, chronic kidney disease * *p* < 0.05.

**Table 4 brainsci-15-00455-t004:** Comparison of least-squares mean values from the multivariable analysis model of the two groups.

Outcome	Variable	Group A-1			Group B-1			P for Interaction
		Coefficient	95% CI	*p*-Value	Coefficient	95% CI	*p*-Value	
GCS score	Time							0.120
	Baseline	Reference			Reference			
	First period	3.095	2.041, 4.148	<0.001	1.565	0.111, 3.020	0.035	
	Second period	4.702	3.648, 5.756	<0.001	3.032	1.578, 4.486	<0.001	
sMRCS score	Time							0.004 *
	Baseline	Reference			Reference			
	First period	3.087	2.165, 4.001	<0.001	2.033	0.759, 3.308	0.002	
	Second period	5.944	5.022, 6.867	<0.001	3.300	2.026, 4.574	<0.001	

CI, confidence interval; GCS, Glasgow coma scale; sMRCS, sum of the Medical Research Council scale, * *p* < 0.05.

## Data Availability

The data that support the findings of this study are available upon reasonable request from the corresponding author.

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
