# Peer review of "Effectiveness of Intracerebral Hemorrhage Aspiration with Catheter Insertion: Impact on Hematoma Volume and Symptom Improvement"

_brainsci, 2025, doi:10.3390/brainsci15050455_

Round 1

Reviewer 1 Report

Comments and Suggestions for Authors

Study Limitations and Recommendations for Improvement

  1. Small Sample Size: The limited sample size may have reduced the statistical power of the findings. Future studies should include larger cohorts to enhance the reliability of conclusions.

  2. Retrospective Design: The retrospective nature of this study carries inherent risks of bias. We recommend prospective randomized controlled trials in future research to minimize potential biases.

  3. Lack of Long-term Follow-up: The study only assessed short-term outcomes without evaluating long-term functional recovery or quality of life changes. Incorporating long-term follow-up data would provide a more comprehensive assessment of treatment efficacy.

  4. Non-standardized Surgical Procedures: The specific intraoperative suction techniques (including depth, range, and pressure parameters) were not thoroughly described, which may affect result reproducibility. Future studies should establish clear procedural standards.

Specific Recommendations for Authors:

  1. Provide detailed descriptions of:

    • Standardized intraoperative suction protocols
    • Potential risks associated with the procedure
    • Incidence of postoperative rebleeding
  2. Include either:

    • Subgroup analyses, or
    • Long-term follow-up data
  3. Expand the discussion to address:

    • Clinical limitations of the findings
    • Directions for future research
  4. Increase the sample size in subsequent studies

  5. Address the rationale for performing surgery within 4 hours of onset in some patients, given existing literature suggesting higher rebleeding risks when operating within 6 hours of intracerebral hemorrhage.

Final Recommendation: While this study demonstrates clinical significance and publication potential, the manuscript requires revision addressing these concerns. A final publication decision should be made after reviewing the revised results.

Author Response

Specific Recommendations for Authors:

  1. Provide detailed descriptions of:
    • Standardized intraoperative suction protocols
    • Potential risks associated with the procedure
    • Incidence of postoperative rebleeding

Response -> The detailed protocols for both aspiration and irrigation have been described. The aspiration volume was determined based on measurements of initial and post-procedural intracranial pressure (ICP), and the exact pressure values were added. We acknowledge that manipulation during the procedure may pose a risk of rebleeding. To minimize this risk during irrigation, the pressure was controlled at 50 mmHg, as stated in the manuscript. Notably, no cases required reoperation due to postoperative rebleeding in this study.

  1. Include either:
    • Subgroup analyses, or
    • Long-term follow-up data

Response -> sMRSC scores were analyzed in the 75% of patients who were followed up at the outpatient clinic. Due to the limited sample size, only general trends could be observed, and this limitation has been additionally addressed in the manuscript.

  1. Expand the discussion to address:
    • Clinical limitations of the findings
    • Directions for future research

Response -> A forward-looking perspective has been added.

  1. Increase the sample size in subsequent studies

Response -> We acknowledge the small sample size; however, we believe it is meaningful that statistically significant P-values were observed for both postoperative imaging differences and clinical symptoms, even within this limited cohort.

  1. Address the rationale for performing surgery within 4 hours of onset in some patients, given existing literature suggesting higher rebleeding risks when operating within 6 hours of intracerebral hemorrhage.

Response -> The rationale for performing early surgery was described, along with practical tips to mitigate the risk of rebleeding. I agree that early surgical intervention may increase the risk of rebleeding. However, this study argues that—aside from the issue of rebleeding—early surgery is more beneficial for functional recovery.

Final Recommendation: While this study demonstrates clinical significance and publication potential, the manuscript requires revision addressing these concerns. A final publication decision should be made after reviewing the revised results.

Reviewer 2 Report

Comments and Suggestions for Authors

Dear Authors,

thank you very much for this study a the presentation of benefits of surgical ICH treatment. I have following questions. How many patients were generally treated with ICH at your institution and how many % of those patients underwent catheter procedure? Do you use other methods as surgical evacuation without endoscope as well? Is it possible to compare the cohorts? Did you found some complications during the treatments? Can you provide the indication process, which patients were selected for the treatment and which not and why? It is sometimes difficult to follow the different groups is it possible to make a diagram or figure describing the different treatment paths? 

Author Response

How many patients were generally treated with ICH at your institution and how many % of those patients underwent catheter procedure?

Response -> We identified patients with spontaneous ICH who were admitted to the Department of Neurosurgery between 2020 and 2024 and categorized them according to the surgical methods.

Do you use other methods as surgical evacuation without endoscope as well?

Response -> In cases where the ICH was located close to the brain cortex, surgical evacuation using a microscope was often performed. As mentioned above, this information has been added as Figure 1. Patients who underwent surgery using a microscope were categorized under the craniotomy + craniectomy group, and the proportion of these cases was compared accordingly.

Is it possible to compare the cohorts?

Response -> Recently reported randomized controlled trials (RCTs) have been added, demonstrating that minimally invasive surgery is more effective than craniotomy or conservative care. This also provides the rationale for comparing the catheter insertion group with the aspiration group in our study.

Did you found some complications during the treatments?

Response -> Although no treatment-related complications were observed, the potential risks were acknowledged. I agree that the risk of rebleeding may increase with greater manipulation.

Can you provide the indication process, which patients were selected for the treatment and which not and why?

Response -> We described the indications for surgical intervention among patients with total ICH. Although the classifications were not based on strict criteria, there were reasons for selecting different surgical methods depending on each case.

It is sometimes difficult to follow the different groups is it possible to make a diagram or figure describing the different treatment paths? 

Response -> The information you mentioned has been summarized and presented in Figure 1. Agreeing that it is an important point, we reanalyzed the data.

Round 2

Reviewer 2 Report

Comments and Suggestions for Authors

Congratulation!